# Is there an association between disease ignorance and self-rated health? The HUNT Study, a cross-sectional survey

Pål Jørgensen,[1] Arnulf Langhammer,[2] Steinar Krokstad,[2] Siri Forsmo[1]

[1]Department of Public Health and General Practice, Norwegian University of Science and Technology, Trondheim, Norway
[2]Department of Public Health and General Practice, HUNT Research Centre, Norwegian University of Science and Technology, Levanger, Norway

**Correspondence to**
Dr Pål Jørgensen;
pal.jorgensen@ntnu.no

## ABSTRACT

**Objective:** To explore whether awareness versus unawareness of thyroid dysfunction, diabetes mellitus or hypertension is associated with self-rated health.

**Design:** Large-scale, cross-sectional population-based study. The association between thyroid function, diabetes mellitus and blood pressure and self-rated health was explored by multiple logistic regression analysis.

**Setting:** The second survey of the Nord-Trøndelag Health Study, HUNT2, 1995–1997.

**Participants:** 33 734 persons aged 40–70 years.

**Primary outcome measures:** Logistic regression was used to estimate ORs for good self-rated health as a function of thyroid status, diabetes mellitus status and blood pressure status.

**Results:** Persons aware of their hypothyroidism, diabetes mellitus or hypertension reported poorer self-rated health than individuals without such conditions. Women with unknown and subclinical hypothyroidism reported better self-rated health than women with normal thyroid status. In women and men, unknown and probable diabetes as well as unknown mild/moderate hypertension was not associated with poorer health. Furthermore, persons with unknown severe hypertension reported better health than normotensive persons.

**Conclusions:** People with undiagnosed but prevalent hypothyroidism, diabetes mellitus and hypertension often have good self-rated health, while when aware of their diagnoses, they report reduced self-rated health. Use of screening, more sensitive tests and widened diagnostic criteria might have a negative effect on perceived health in the population.

## INTRODUCTION

Guidelines for prevention and treatment have been developed for most high-prevalent diseases in western countries aiming for a reduction of morbidity and mortality by interventions mainly in primary healthcare (PHC).

In society, there seems to be an increasing conviction of achievable zero vision regarding risks and diseases. Part of the strategy is to

## Strengths and limitations of this study

- Sample from a large-scale general population.
- High-prevalent diseases under study; ensuring statistical power in subgroup analyses.
- Study mainly based on self-reported data.
- Cross-sectional study; susceptibility to confounding and impossibility to assume causal relationships.

detect risk factors and prediseases in even earlier stages. From a secondary or tertiary healthcare level, this might seem reasonable since intervention on many individuals with specific risk factors presumably can prevent or delay disease or progression of disease. Furthermore, health authorities and hospital clinicians regularly raise concern of the lack of detection of risk factors, of subclinical conditions and of achieving treatment goals.[1–7] Norwegian studies have shown that guidelines are often difficult to implement and adhere to in PHC.[8 9] According to guidelines, most individuals would be defined as at risk and resources needed to handle this appropriately could destabilise the entire healthcare system.[10–12] An American review pointed out knowledge, attitude and behaviour as barriers to physician adherence to clinical guidelines.[13] In an already complex and busy PHC setting, one might expect that resources used for disease prevention and case finding have to compete with resources for handling acknowledged disease. Also, physicians might want to avoid increasing disease-related burden for patients, in line with the old wisdom: "primum non nocere".[14] Thus, the risk and disease zero visions in society and among politicians are seldom shared by PHC professionals.

When guidelines, mainly based on research from high-risk hospital populations, are applied on low-risk populations in PHC, more healthy individuals are identified as being at risk or are given diagnoses. Also, the widened inclusion criteria for diagnoses in general and

use of more sensitive tests contribute to define more individuals at risk or as unhealthy.[15] The possible undesirable outcomes of such strategies remain unclear.

Self-rated health (SRH) is a valid and widely used measure of general health in epidemiological research.[16–19] It is associated with several clinical conditions often seen in PHC, with recovery,[20–23] and is found to predict morbidity, sick leave and disability pension,[24 25] as well as mortality.[26–28] The majority of studies describing an association between labelling of disease per se and SRH have focused on arterial hypertension.[29–32] However, one study has indicated reduced SRH among individuals labelled with thyroid dysfunctions.[33]

The aim of this study was to investigate whether persons' awareness versus unawareness of thyroid dysfunction, diabetes mellitus or hypertension was associated with their SRH, as reported in a population-based health study in Norway.

## METHODS
### Study population
The data sample in this study stems from the second wave of the Nord-Trøndelag Health Study (HUNT2) conducted in 1995–1997 in the county of Nord-Trøndelag, Norway. All individuals aged 20 years and older, living in the county, were invited (94 194 individuals). In all, 66.7% of men (n=30 860) and 75.5% of women (n=35 280) participated. The survey consisted of questionnaires and measurements, and has previously been described in detail.[34 35] In our study, we included answers from the main questionnaire and the baseline measurements for persons aged 40–70 years. The age span was chosen because thyroid stimulating hormone (TSH) was analysed in all women and in 50% of men at this age, of a rather low disease burden in people younger than 40 years and of a lower attendance rate under and above this age span. A total of 24 950 individuals had TSH measurements and answered thyroid questions and thus were eligible for analysis on thyroid dysfunction, while in the analysis of diabetes mellitus and blood pressure (BP), 33 734 individuals were included in all.

### Self-rated health
The first question in the main questionnaire in HUNT2, answered before attending the examination stations, was "How is your health at the moment?", with answer alternatives 'very good', 'good', 'not so good' and 'poor'. This short version of SRH measure is shown to be a valid predictor of mortality.[18 28 36 37] We dichotomised the answers into 'good' (very good, good) and 'poor' (not so good, poor). Dichotomisation of multinomial SRH is commonly performed, and has been validated by Manor et al.[38]

### Thyroid function
The participants answered questions on the history of hypothyroidism and hyperthyroidism, goitre, other thyroid diseases and treatment with thyroxin, radioiodine, surgery or thyreostatic medication.

Serum TSH and free T4 were analysed at the hormone laboratory, Aker University Hospital, Norway. The laboratory reference value for TSH, as defined prior to the survey, was 0.2–4.5 mU/L and for free T4 8–20 pmol/L. If TSH was <0.2 or >4 mU/L, and/or if the participant reported any thyroid disease, serum free T4 was also measured.[39]

Individuals reporting no previous thyroid disease and having TSH within the reference range were categorised as 'no thyroid disease' and chosen as the reference category. No previous thyroid disease combined with TSH >4.5 mU/L and free T4 <8 pmol/L was defined as unknown hypothyroidism. No previous thyroid disease combined with TSH >4.5 mU/L and free T4 8–20 pmol/L was defined as subclinical hypothyroidism. Individuals reporting hypothyroidism and use of thyroxin were classified as having known hypothyroidism, regardless of the TSH and T4 levels. Affirmative answers to other thyroid-related questions or measures outside the reference range in the remainders were classified as other thyroid dysfunction.

### Diabetes mellitus
Diabetes mellitus was assessed through self-report and blood samples. Serum glucose was analysed at Levanger Hospital, Norway. Those reporting no diabetes and having normal glucose levels (<5.5 mmol/L) were classified as 'no diabetes' and were chosen as the reference category. No self-reported diabetes and non-fasting glucose >11 mmol/L was categorised as unknown diabetes, whereas no diabetes and non-fasting glucose 5.5–11 mmol/L was categorised as probable diabetes. Self-reported diabetes was classified as known diabetes regardless of the glucose level.

### Blood pressure
In the questionnaire, participants were asked about the doctor's advice after the latest BP measurement prior to participation in HUNT. The answer categories were: 'no follow-up and no medication necessary', 'recommended follow-up examination but not to take medicine', 'start or continue taking medicine for high blood pressure' or 'never measured'. At HUNT2, mean systolic and mean diastolic arterial BP of measurement 2 and 3 was categorised into normal (systolic (s) BP<140 mm Hg and diastolic (d) BP<90 mm Hg), mild hypertension (sBP 140–159 mm Hg and dBP<100 mm Hg or sBP<60 mm Hg and dBP 90–99 mm Hg), moderate hypertension (sBP 160–179 mm Hg and dBP<109 mm Hg or sBP<180 mm Hg and dBP 100–109 mm Hg) and severe hypertension (sBP>180 mm Hg or dBP>110 mm Hg). We constructed a new variable to define normotensive (reference), unknown mild and moderate hypertensive, unknown severe hypertensive and known hypertensive persons on the basis of self-report and measures.

## Statistical analysis

The descriptive analyses of the study population were stratified by gender, and we used $\chi^2$ tests to examine any difference in proportions of SRH between the independent variables. Gender-stratified multiple logistic regression was used to estimate OR with 95% CI for good SRH, as a function of thyroid status, diabetes mellitus status and BP status, in separate unadjusted, age-adjusted and multiadjusted analyses for each condition.

Age, smoking, alcohol consumption, body mass index (BMI), working and educational status, and self-reported limiting long-term illness or injury are associated with SRH[40] and the diseases under study, but most likely not affected by SRH or the diseases. Hence, these variables were included, a priori, as confounders in the models. Age was categorised into age groups 40–49, 50–59 and 60–70 years. Smoking status was categorised into never smoked daily, previous daily smoker and current daily smoker. Alcohol units (AU) were defined as number of glasses of wine, beer or liquor. Those reporting to be teetotallers or to have alcohol intake less than four times a month or less than 7 AU/2 weeks were categorised as low consumers, those reporting drinking five to eight times a month or 8–14 AU/2 weeks as moderate consumers and those drinking more often than eight times a month or more than 14 AU as heavy consumers. BMI ($kg/m^2$) was calculated of measured height and weight and categorised according to the WHO definition: underweight ($18.5\ kg/m^2$), normal weight ($18.5$–$24.9\ kg/m^2$), overweight ($25$–$29.9\ kg/m^2$) and obese ($>30\ kg/m^2$). People reporting paid-employed or self-employed work were classified as working, otherwise as not working. Educational level was categorised into <10, 10–12 and >12 years. We chose an affirmative answer to the question "Do you suffer from any long-term illness or injury (at least one year) of a physical or psychological nature that impairs your functioning in your everyday life?" to represent all relevant chronic medical conditions that could confound the results.

To examine whether the association of the three disease statuses with SRH differed by categories of the other independent variables, we used likelihood ratio tests with p value for statistical interaction. We tested for multicollinearity between the independent variables by linear regression. Areas under the receiver operating characteristic curves (AUC) were calculated to evaluate the performance of the logistic regression models.

In an additional analysis, the association between SRH and having had one or more medical consultations during the past year was investigated by logistic regression models, stratified by gender, in the total study population and after exclusion of persons with diagnoses under study.

All analyses were performed with IBM SPSS Statistics V.20 for Windows.

All participants signed a written informed consent.

## RESULTS

In all age categories, a higher proportion of men than women reported good SRH (p<0.001), and in both sexes the proportion reporting good SRH declined by age (p<0.002; table 1). The proportion reporting good SRH was lower in overweight, obese and underweight women than in normal weight women (p<0.001). In men, the proportion reporting good SRH declined between the normal weight group and the overweight, obese and underweight groups (p<0.001). In previous and current female smokers, a lower proportion reported good SRH compared with non-smokers (p<0.001). In men, the proportion reporting good SRH was higher among non-smokers than among previous and current smokers (p<0.001), whereas there was no difference between previous and current smokers. The proportion reporting good SRH was higher in moderate and heavy alcohol consumers than in low consumers, increased with education and was higher for participants in paid work (p<0.001). Underweight men and persons with 'any long-term impairment' had the lowest proportion reporting good SRH of all groups. The proportion reporting good SRH in the overall HUNT2 Study population differed from the proportions reported in persons without thyroid disease and normotensive persons (p<0.01), but not from persons without diabetes mellitus. However, the absolute differences were small.

In all fully adjusted regression models, the AUC was 0.80. Unadjusted, the AUC ranged from 0.61 to 0.63.

There was no multicollinearity between the independent variables.

### Thyroid function

Thyroid dysfunctions, known and unknown, were more often observed among women than men (p<0.001; table 1). Women with known hypothyroidism had a lower OR of reporting good SRH than women without thyroid disease in the adjusted analyses. However, women with unknown or subclinical hypothyroidism had an 84% and 48% higher OR, respectively, of reporting good SRH compared with the OR of women without thyroid disease (table 2). The association between thyroid function and SRH was basically unchanged after inclusion of confounder variables. Corresponding, but non-significant, associations were found among men.

### Diabetes mellitus

The prevalence of unknown, probable and known diabetes was slightly higher in men than in women (p<0.001; table 1). Women with known diabetes mellitus had a lower OR of good SRH than those without diabetes in the adjusted analyses, whereas in women with unknown or possible diabetes mellitus, the ORs of good SRH were similar to the ORs among persons without diabetes in the adjusted analyses (table 2). In the adjusted analyses, the association between unknown and probable diabetes mellitus with poor SRH, found in the

**Table 1** Good SRH by sex and characteristics of the study population

| Study population (n=33 734) | Women (n=17 514) | | | Men (n=16 220) | | |
|---|---|---|---|---|---|---|
| | n | Per cent | SRH good (%) | n | Per cent | SRH good (%) |
| Age group (years) | | | | | | |
| 40–50 | 7058 | 40.3 | 77.6 | 6517 | 40.2 | 81.7 |
| 51–60 | 5709 | 32.6 | 64.4 | 5328 | 32.8 | 72.1 |
| 61–70 | 4747 | 27.1 | 56.6 | 4375 | 27.0 | 59.6 |
| BMI (kg/m$^2$) | | | | | | |
| <18.5 | 6821 | 0.6 | 58.4 | 4804 | 0.2 | 42.9 |
| 18.5–24.9 | 7026 | 39.1 | 72.8 | 8694 | 29.7 | 75.3 |
| 25.0–29.9 | 3489 | 40.3 | 68.5 | 2635 | 53.8 | 73.6 |
| >30 (0.4% missing) | 104 | 20.0 | 56.5 | 35 | 16.3 | 64.8 |
| Smoking status | | | | | | |
| Never smoked daily | 6973 | 40.1 | 70.3 | 4849 | 30.1 | 80.0 |
| Previous daily smoker | 4651 | 26.7 | 68.0 | 6105 | 37.9 | 69.8 |
| Daily smoker (0.7% missing) | 5763 | 33.1 | 64.3 | 5172 | 32.0 | 69.1 |
| Alcohol use | | | | | | |
| None to low intake | 14 888 | 88.7 | 66.7 | 11 488 | 73.2 | 71.0 |
| Moderate intake | 1471 | 8.8 | 76.8 | 2839 | 18.1 | 78.7 |
| High intake (3.7% missing) | 430 | 2.6 | 76.3 | 1375 | 8.8 | 78.3 |
| Educational level (years) | | | | | | |
| <10 | 8133 | 48.5 | 60.5 | 5711 | 36.5 | 63.1 |
| 10–12 | 5682 | 33.9 | 72.8 | 6615 | 42.2 | 76.0 |
| >12 (3.8% missing) | 2962 | 17.7 | 80.0 | 3339 | 21.3 | 84.7 |
| Employed | | | | | | |
| Yes | 11 539 | 67.2 | 77.1 | 12 309 | 77.1 | 79.9 |
| No (1.8% missing) | 5627 | 32.8 | 49.0 | 3665 | 22.9 | 49.2 |
| Any long-term impairment | | | | | | |
| No | 10 348 | 59.1 | 85.9 | 9912 | 63.0 | 88.7 |
| Yes (4.1% missing) | 6275 | 35.8 | 39.8 | 5829 | 37.0 | 46.0 |
| Thyroid function | | | | | | |
| No thyroid disease | 14 373 | 86.2 | 68.7 | 7804 | 94.3 | 72.1 |
| Unknown hypothyroidism | 107 | 0.6 | 78.5 | 16 | 0.2 | 68.8 |
| Subclinical hypothyroidism | 466 | 2.8 | 75.0 | 150 | 1.8 | 66.4 |
| Known hypothyroidism | 858 | 5.1 | 48.9 | 124 | 1.5 | 58.1 |
| Other thyroid dysfunction (1.3% missing) | 872 | 5.2 | 61.4 | 180 | 2.2 | 56.2 |
| Diabetes mellitus | | | | | | |
| No diabetes | 11 279 | 64.6 | 69.4 | 9196 | 56.9 | 74.6 |
| Unknown diabetes | 46 | 0.3 | 52.2 | 88 | 0.5 | 67.8 |
| Probable diabetes | 5728 | 32.8 | 65.9 | 6392 | 39.6 | 71.5 |
| Known diabetes (0.4% missing) | 395 | 2.3 | 43.1 | 482 | 3.0 | 49.5 |
| Blood pressure status | | | | | | |
| Normotensive | 9135 | 52.6 | 72.4 | 6858 | 42.7 | 77.5 |
| Unknown mild/moderate hypertension | 4473 | 25.8 | 67.9 | 5343 | 33.3 | 74.7 |
| Unknown severe hypertension | 403 | 2.4 | 70.7 | 349 | 2.2 | 73.8 |
| Known hypertension (0.9% missing) | 3327 | 19.2 | 54.0 | 3498 | 21.8 | 59.6 |
| Overall study population, HUNT2 | 34 332 | 97.3 | 70.3 | 30 378 | 98.4 | 74.9 |

BMI, body mass index; HUNT, Nord-Trøndelag Health Study; SRH, self-rated health.

crude analyses, disappeared when age was included in the model in women and men, but also by inclusion of working status alone in women. In men, the association of diabetes status with SRH differed by levels of education. Among men without higher education (12 years or less), the ORs of good SRH were as in the main effect model (table 2). However, in men with higher education, the ORs of good SRH were barely significantly lower among men with unknown diabetes (OR 0.29 (95% CI 0.09 to 1.00)) and among men with possible diabetes (OR 0.79 (95% CI 0.63 to 1.00)) compared with men without diabetes. Among men with known diabetes in this stratum, the OR of good SRH was 0.31 (95% CI 0.17 to 0.54) compared with men without diabetes.

**Blood pressure**

The prevalence of unknown mild and moderate hypertension was higher in men than in women (p<0.001). Unknown severe hypertension and known hypertension

**Table 2** The association between self-rated health and thyroid function, diabetes mellitus and blood pressure

| | Women, OR (95% CI) | | | | Men, OR (95% CI) | | | |
|---|---|---|---|---|---|---|---|---|
| | n | Crude | Age adjusted | Multiple adjusted | n | Crude | Age adjusted | Multiple adjusted |
| Thyroid function | | | | | | | | |
| No thyroid dysfunction | 12 476 | 1.00 | 1.00 | 1.00 | 7045 | 1.00 | 1.00 | 1.00 |
| Unknown hypothyroidism | 92 | 1.66 (1.05 to 2.64) | 2.00 (1.25 to 3.19) | 1.84 (1.02 to 3.33) | 14 | 0.85 (0.30 to 2.45) | 0.92 (0.31 to 2.72) | 1.28 (0.35 to 4.65) |
| Subclinical hypothyroidism | 394 | 1.37 (1.10 to 1.69) | 1.49 (1.20 to 1.85) | 1.48 (1.13 to 1.94) | 128 | 0.77 (0.54 to 1.08) | 0.88 (0.62 to 1.24) | 1.13 (0.72 to 1.76) |
| Known hypothyroidism | 718 | 0.44 (0.38 to 0.50) | 0.48 (0.41 to 0.55) | 0.49 (0.41 to 0.59) | 107 | 0.54 (0.37 to 0.77) | 0.63 (0.43 to 0.91) | 0.69 (0.44 to 1.09) |
| Other thyroid dysfunction | 729 | 0.72 (0.63 to 0.83) | 0.77 (0.67 to 0.89) | 0.77 (0.65 to 0.93) | 163 | 0.50 (0.37 to 0.67) | 0.52 (0.38 to 0.71) | 0.58 (0.41 to 0.84) |
| Diabetes mellitus | | | | | | | | |
| No diabetes | 9946 | 1.00 | 1.00 | 1.00 | 8464 | 1.00 | 1.00 | 1.00 |
| Unknown diabetes | 39 | 0.48 (0.27 to 0.86) | 0.61 (0.34 to 1.10) | 0.66 (0.32 to 1.37) | 82 | 0.72 (0.46 to 1.13) | 0.88 (0.55 to 1.39) | 1.03 (0.59 to 1.81) |
| Probable diabetes | 4797 | 0.85 (0.80 to 0.91) | 0.94 (0.88 to 1.01) | 1.01 (0.92 to 1.10) | 5759 | 0.86 (0.80 to 0.92) | 0.94 (0.88 to 1.01) | 0.99 (0.91 to 1.08) |
| Known diabetes | 318 | 0.33 (0.27 to 0.41) | 0.42 (0.34 to 0.51) | 0.53 (0.41 to 0.69) | 395 | 0.33 (0.28 to 0.40) | 0.42 (0.35 to 0.51) | 0.55 (0.43 to 0.70) |
| Blood pressure | | | | | | | | |
| No hypertension | 8180 | 1.00 | 1.00 | 1.00 | 6378 | 1.00 | 1.00 | 1.00 |
| Unknown mild/moderate hypertension | 3787 | 0.81 (0.75 to 0.87) | 1.01 (0.93 to 1.09) | 1.10 (0.99 to 1.22) | 4837 | 0.86 (0.79 to 0.93) | 1.01 (0.92 to 1.10) | 1.01 (0.92 to 1.12) |
| Unknown severe hypertension | 325 | 0.92 (0.74 to 1.15) | 1.34 (1.08 to 1.68) | 1.52 (1.14 to 2.02) | 308 | 0.85 (0.77 to 1.08) | 1.27 (0.99 to 1.62) | 1.48 (1.09 to 2.02) |
| Known hypertension | 2745 | 0.45 (0.41 to 0.49) | 0.59 (0.54 to 0.65) | 0.69 (0.61 to 0.77) | 3118 | 0.43 (0.39 to 0.47) | 0.54 (0.50 to 0.60) | 0.64 (0.57 to 0.72) |

OR of good self-rated health and 95% CIs, crude, age adjusted, and adjusted for age, other long-term illness or injury that impairs function in everyday life, smoking habits, alcohol use, educational level, work status and body mass index. Cases with missing data were excluded from the analyses.

were equally distributed between women and men (table 1). Women with known hypertension had a lower OR of reporting good SRH than normotensive women in the adjusted analyses. The figures were similar in men (table 2). In contrast, compared with normotensive women, those with unknown severe hypertension had a 52% higher OR of reporting good SRH, with similar figures in men. Persons with unknown mild and moderate hypertension reported good SRH, similar to the normotensive ones. Adjusted for age, the association between unknown mild and moderate hypertension and poor SRH disappeared simultaneously; unknown severe hypertension became associated with good SRH in women. In men, age had to be added along with either education or working status to achieve the latter association.

### Additional analyses

Women with poor SRH had more than six times the OR of those with good SRH to have had a medical consultation during the last year; OR 6.29 (95% CI 5.47 to 7.22). For men, the corresponding OR was 5.53 (95% CI 4.86 to 6.29). After exclusion of persons with diagnosed thyroid disease, diabetes mellitus, known hypertension and 'any long-term disease', the corresponding OR was 3.71 (95% CI 2.90 to 4.73), with similar figures in men.

### DISCUSSION

This large population-based study showed that persons with known thyroid dysfunction, diabetes mellitus and hypertension were less likely to report good SRH than those without such conditions. Less expected, persons with unknown and subclinical hypothyroidism were more likely to report good SRH than those without thyroid disease. Similarly, those with unknown severe hypertension were more likely to report good SRH, compared with persons with normal BP. In general, persons with unknown diabetes and unknown mild/moderate hypertension reported good health, just like the reference group.

In general, of the confounders, age seemed to influence the association between disease status and SRH most when adjusted for. Age was found to explain the association of poor SRH with unknown and probable diabetes, and with unknown mild and moderate hypertension. In women, age even contributed to an association between unknown severe hypertension and good SRH. There seemed to be a linear decrease by age categories in the association with good SRH. The way age is known to be related to disease and SRH makes these findings reasonable.

Although a qualitative and quantitative measure of association, the estimated ORs in our study will, if interpreted as relative risks, overestimate the case, because the prevalence of good SRH is high in all groups.[41]

The main strengths of this study were: the number of participants, that the diseases we studied were high prevalent and that we could assume representativeness to the general population regarding the variables included. It is known that individuals with a high burden of symptoms are less likely to attend surveys, but so far there has been little evidence that non-participation on this basis introduces substantial bias in associational studies.[42] We studied diseases with a generally low symptom burden and expect selection bias to be negligible.

The main limitation was possibly that we relied on self-reported data on dependent and independent variables. The validity of self-reported measures relevant for this study is questionable. However, in case of any misclassification, it should be non-differential, thus should only cause an underestimation of the associations found.

There is always a possibility of residual confounding in non-randomised study designs, and owing to the observational, cross-sectional design, we cannot assume a causal relationship.

Bias due to differential detection of disease in the study population could lead to a type 1 error. Hypothyroidism and diabetes mellitus are often associated with vague symptoms such as tiredness and weakness. These symptoms are strongly associated with reduced SRH.[43] It is most likely that presenting such symptoms for the general practitioner (GP) would result in measurements of TSH, free T4 and serum glucose, thus revealing any related dysfunction. Low self-perceived health increases the probability of visiting a GP.[44] Social security covers most of the costs related to clinical measurements and blood sampling in Norway; thus, we expect GPs to have a low threshold to measure thyroid function, BP or glucose levels. People who do not consult their GP will not have diseases with mild or no symptoms diagnosed, and their personality could be characterised by a less worrying and more optimistic attitude to health being reflected in better SRH.[45]

On the other hand, the association between known disease and poor SRH could in fact be explained physiologically due to the pathological effect of disease. Lack of a corresponding association between ignored, but prevalent, disease and poor SRH is not in line with this hypothesis, raising the question of a possible adverse effect of disease labelling. However, confounding by severity of disease cannot be ruled out.

Owing to the low number of persons having unknown hypothyroidism and diabetes mellitus, the results should be interpreted with caution. On the other hand, there were high numbers in the subclinical and probable groups, and analyses of these groups did not show any associations with poor SRH, also raising questions of a possible disease labelling effect.

The fact that the HUNT2 survey was carried out nearly 20 years ago raises a question of generalisability to today's population. Stability of SRH over time has not been investigated in our study population, although it has been investigated among adolescents.[46] We do not expect the association between low burden and subclinical disease

with SRH to be time dependent to the extent that it would change our results considerably. Neither do we expect the changes in prevalence of most explanatory variables to influence the associations found.

Consistent findings regarding unknown, subclinical and probable disease versus known disease could indicate a potential adverse effect of disease labelling. Although early detection of disease is protective on morbidity and mortality for many diseases, low SRH is also shown to be associated with morbidity and mortality.[25–27] This is an important aspect in the debate of presymptomatic case finding.

The BMJ's Too Much Medicine campaign aims to highlight the threat to human health posed by overdiagnosis and the waste of resources on unnecessary care (http://www.bmj.com/too-much-medicine). American data have shown that the majority of all healthcare includes preference-sensitive and supply-sensitive services. The extent of these services varies greatly without necessarily leading to better health.[47 48] A great deal of activity in such services is based on identifying subclinical disease. The negative health effect this might have, modulated through reduced SRH, can explain why population mortality is not reduced in areas with a high frequency of diagnoses.

Public and academic debates are often characterised by the conviction that all medical treatment is efficient. Wennberg[49] showed that a relatively small proportion of all medical treatment is indisputably good for health. The concern for unrevealed risk factors and subclinical conditions might lead to unnecessary costly healthcare interventions, increase supply-sensitive services without positive health effect and have a negative influence on people's general and self-perceived health causing more harm than good.[15]

Of ethical reasons, the possible causal effect of disease labelling on SRH is impossible to assess in a randomised controlled trial. Our study emphasises the need for more prospective research to investigate the potential health effects of disease labelling and early diagnosis.

## CONCLUSION

Our data suggest that early identification of disease may imply a negative effect on SRH, and to the extent that SRH has been associated with greater mortality, this may lead to harm. However, as it is also known that diseases such as diabetes and hypertension also lead to increased mortality when detected after cardiovascular and metabolic complications have developed, it remains to be seen whether an early identification or a late detection strategy would provide optimal health for the population.

**Acknowledgements** The Nord-Trøndelag Health Study (The HUNT Study) is a collaboration between the HUNT Research Centre (Faculty of Medicine, Norwegian University of Science and Technology NTNU), the Nord-Trøndelag County Council, the Central Norway Health Authority and the Norwegian Institute of Public Health. The authors would like to thank Bjørn Olav Åsvold for his contribution in planning the thyroid part of the study.

**Contributors** SF, AL and SK have been active supervisors in the study conception, design, conduct, interpretation and reporting. PJ analysed the data and drafted the manuscript. Critical revisions were carried out by all supervisors and all authors approved the final version of the manuscript. SF, AL, SK and PJ are joint guarantors.

**Funding** PJ received a PhD grant, funded by the Norwegian University of Science and Technology, NTNU.

**Competing interests** None.

**Ethics approval** The study was approved by the Regional Committee for Medical Research. All participants signed a written informed consent.

**Provenance and peer review** Not commissioned; externally peer reviewed.

**Data sharing statement** No additional data are available.

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
