## [Reviewer comments · BMJ Open]

Some articles will have been accepted based in part or entirely on reviews undertaken for other BMJ Group journals. These will be reproduced where possible.

ARTICLE DETAILS

TITLE (PROVISIONAL)	Is there an association between disease ignorance and self-rated health? The HUNT Study, a cross-sectional survey
AUTHORS	Jørgensen, Pål; Langhammer, Arnulf; Krokstad, Steinar; Forsmo, Siri

VERSION 1 - REVIEW

REVIEWER	Prof. Paul Froom MD School of Public Health\ Sackler Medical School Tel Aviv University Tel Aviv, Israel
REVIEW RETURNED	05-Mar-2014

GENERAL COMMENTS	Limitations not discussed was the fact that 1. Data collection was around 20 years ago2. It appears that the division into two categories of SRH was done after the onset of analysis. The area under the curve AUC was not calculated. This is needed in order to determine how good the logistical model was in predicting SRH and possibly done with and without the major predictor variables. The effect of using more categories for SRH needs to be reported, or at least mentioned. A final model of all variables that add significantly to the model should be presented. This was discussed , but isn't it more likely that patients who felt better were less likely to visit the physician, as evident by SRH higher than the reference group. The adjustment that led to changes in the crude rates, needs to be addressed, and the reasons explored. Which confounders led to those changes? Were there any interactions with those variables?
--

REVIEWER	Amy Hubschmann, MD University of Colorado School of Medicine, United States of America
REVIEW RETURNED	29-Mar-2014

GENERAL COMMENTS	Summary: Self-rated Health (SRH) is important because it is related to morbidity and mortality on a population level. Certain diseases
--

such as arterial hypertension and hypothyroidism have been associated with lower self-rated health. The authors assessed the relationship between self-rated health among those with known and unknown arterial hypertension, hypothyroidism, and diabetes mellitus. This was a population-based study from a single county in Norway with very good response rates from the county residents (67% response rate for men; 75% response rate for women). This study included only a subset of county residents who had lab results for thyroid dysfunction (n = 24,950) and tests for hypertension and diabetes (n = 33,734). Disease states were categorized as known or unknown disease for thyroid, diabetes, and hypertension on the basis of disease self-report (all), lab results (thyroid, diabetes), and blood pressure results (hypertension)

Major criticisms:

1. The authors do not defend the validity of the SRH question used – “How is your health at the moment?” – has this specific type of SRH question been validated to relate to mortality and morbidity? A related concern is whether dichotomizing the SRH responses into “good” and “poor” is appropriate rather than considering this as a categorical variable for all 4 responses. I would ask the authors to reference the prior studies that demonstrated the validity of this question and the validity of using the dichotomous responses used in the analysis.
2. We do not know how well the participants with measurements of thyroid, diabetes, and hypertension are representative of the overall study population. This is important from an external validity standpoint. To inform the readers, will you please compare the rates of “good” SRH and known thyroid, diabetes, and hypertension reported by the study population (n = 33,734 and n = 24,950, respectively for dm/htn and thyroid) as compared to the overall study population (n = ~66,000)?
3. I am troubled by the first sentence of your conclusion, “Our data suggest that early identification of disease may imply a negative effect on SRH and thereby eventually cause more harm than good.” Late detection of hypertension and diabetes, in particular, may lead to a cardiovascular event as the first manifestation of disease – that is also harmful! I suggest rewording to note that early identification of disease may imply a negative effect on SRH, to the extent that SRH has been associated with greater mortality, this may lead to poorer health. However, as it is also known that diseases such as diabetes and hypertension also lead to worse mortality when detected after cardiovascular and metabolic complications have developed, it remains to be seen if an early identification or a late detection strategy would provide optimal health for the population.

VERSION 1 – AUTHOR RESPONSE

Reviewer Name Prof. Paul Froom MD

Institution and Country School of Public Health\Sackler Medical School

Tel Aviv University

Tel Aviv, Israel

Please state any competing interests or state 'None declared': None declared

Limitations not discussed was the fact that:

1. Data collection was around 20 years ago.

Response: We agree that this should be discussed, and have revised the discussion section in the paper accordingly.

2. It appears that the division into two categories of SRH was done after the onset of analysis.

Response: Thank you for this relevant comment. The use of four answer categories of the SRH question was decided prior to the first HUNT survey (HUNT1) in 1984-86 with the intention to make participants choose between positive and negative answer. This is sometimes called a “forced choice” method in a bipolar scaling method/Likert scaling. Frequent HUNT papers including this variable have used the dichotomized form [1 2] and this was also the a priori plan for this study. To investigate the effect of using all categories of SRH, ordinal regression analyses would ideally be applied. However, we found that the assumption of proportional odds was violated, thus ordinal regression models would not be valid. In binary regression analyses of cumulative categories of the four SRH answer categories, we observed a cut off in the association with the diseases under study between the level “not so good” and “good”, and argue that it gives some support to the validity of our dichotomization. The area under the curve AUC was not calculated. This is needed in order to determine how good the logistical model was in predicting SRH and possibly done with and without the major predictor variables.

Response: We performed an investigation of the association between SRH and disease statuses, in a cross sectional study, thus we believe prediction analyses are not needed in this setting. However, as a response to this comment we have done ROC analyses to calculate the AUC (not reported in the paper):

Thyroid function:

- For women the AUC increased from 0.62 in the age adjusted ROC analysis, till 0.80 in the fully adjusted analysis, indicating good model performance.
- For men the corresponding figures were an increase from 0.61 till 0.80.

Diabetes mellitus:

- For women the AUC increased from 0.62 in the age adjusted ROC analysis, till 0.80 in the fully adjusted analysis, indication good model performance.
- For men the corresponding figures were exactly the same.

Blood pressure:

- For women the AUC increased from 0.63 in the age adjusted ROC analysis, till 0.80 in the fully adjusted analysis, indicating good model performance.
- For men the corresponding figures were exactly the same.

The effect of using more categories for SRH needs to be reported, or at least mentioned.

Response: We have answered this comment as part of the response to comment no. 2 above.

A final model of all variables that add significantly to the model should be presented.

Response: All variables in the models were selected a priori, based on clinical knowledge, not on statistical significance testing. Our strategy was to adjust for major confounders available in the dataset. Throughout the analyses, we generally found that all the included confounders had statistical significant association with SRH, and the general findings are reported in the results section. After a priori selection, we ran crude, age adjusted, and multiple adjusted binary logistic regression analyses. We have revised table 2 to help enlighten the associations for the readers.

This was discussed, but isn't it more likely that patients who felt better were less likely to visit the

physician, as evident by SRH higher than the reference group.

Response: We agree that this is the most important opposition of the results per se, and plan to investigate this question further in future research.

The adjustment that led to changes in the crude rates, needs to be addressed, and the reasons explored. Which confounders led to those changes? Were there any interactions with those variables?

Response: We have now revised the results and discussion section, as well as table 2, according to this relevant comment. However, in the final, multiple adjusted model, the statistical effect of each variable change dependently on all the other variables included, hence are difficult to interpret. There were no statistical interactions, except between educational level and diabetes status in men; reported in results section.

The paper is well written, but needs a lot of work to convince us that it adds anything to the literature.

Response: We are grateful for the comments and think your contribution has increased the quality of the paper.

Reviewer Name Amy Hubschmann, MD

Institution and Country University of Colorado School of Medicine, United States of America

Please state any competing interests or state 'None declared': None declared

Summary: Self-rated Health (SRH) is important because it is related to morbidity and mortality on a population level. Certain diseases such as arterial hypertension and hypothyroidism have been associated with lower self-rated health. The authors assessed the relationship between self-rated health among those with known and unknown arterial hypertension, hypothyroidism, and diabetes mellitus. This was a population-based study from a single county in Norway with very good response rates from the county residents (67% response rate for men; 75% response rate for women). This study included only a subset of county residents who had lab results for thyroid dysfunction (n = 24,950) and tests for hypertension and diabetes (n = 33,734). Disease states were categorized as known or unknown disease for thyroid, diabetes, and hypertension on the basis of disease self-report (all), lab results (thyroid, diabetes), and blood pressure results (hypertension)

Major criticisms:

1. The authors do not defend the validity of the SRH question used – “How is your health at the moment?” – has this specific type of SRH question been validated to relate to mortality and morbidity? A related concern is whether dichotomizing the SRH responses into “good” and “poor” is appropriate rather than considering this as a categorical variable for all 4 responses. I would ask the authors to reference the prior studies that demonstrated the validity of this question and the validity of using the dichotomous responses used in the analysis.

Response: Thank you for this important comment. Multiple studies have shown that SRH independently relates to morbidity and mortality, and we have now included relevant references in the paper, as requested. To our knowledge, validation of dichotomization of the multinomial SRH measure has only been investigated in one previous study (Manor et. al.); now referenced in our paper. However, as described by Manor et al, SRH is often collapsed into a dichotomous variable when it is used as a dependent variable. We have commented further on this topic as a response to comment no. 2 of the first reviewer.

2. We do not know how well the participants with measurements of thyroid, diabetes, and hypertension are representative of the overall study population. This is important from an external validity standpoint. To inform the readers, will you please compare the rates of “good” SRH and known thyroid, diabetes, and hypertension reported by the study population (n = 33,734 and n = 24,950, respectively for dm/htn and thyroid) as compared to the overall study population (n = ~66,000)?

Response: We have made changes according to this appropriate comment; Table 1 now shows the proportion with good SRH in the overall study population. The test result is also reported early in the results section.

3. I am troubled by the first sentence of your conclusion, “Our data suggest that early identification of

disease may imply a negative effect on SRH and thereby eventually cause more harm than good.” Late detection of hypertension and diabetes, in particular, may lead to a cardiovascular event as the first manifestation of disease – that is also harmful! I suggest rewording to note that early identification of disease may imply a negative effect on SRH, to the extent that SRH has been associated with greater mortality, this may lead to poorer health. However, as it is also known that diseases such as diabetes and hypertension also lead to worse mortality when detected after cardiovascular and metabolic complications have developed, it remains to be seen if an early identification or a late detection strategy would provide optimal health for the population.
 Response: We do agree, and have changed the sentences accordingly.

1. Dalen JD, Huijts T, Krokstad S, et al. Are there educational differences in the association between self-rated health and mortality in Norway? The HUNT Study. Scand J Public Health 2012;40(7):641-7 doi: 10.1177/1403494812459817[published Online First: Epub Date]].
2. Cuypers K, Krokstad S, Holmen TL, et al. Patterns of receptive and creative cultural activities and their association with perceived health, anxiety, depression and satisfaction with life among adults: the HUNT study, Norway. J Epidemiol Community Health 2012;66(8):698-703 doi: DOI 10.1136/jech.2010.113571[published Online First: Epub Date]].

VERSION 2 – REVIEW

REVIEWER	Paul Froom school of Public health University Tel Aviv
REVIEW RETURNED	02-May-2014

GENERAL COMMENTS	First of it would be beneficial to be able to read my previous review of this paper. secondly the regression models should have reported the areas under the curve. thirdly , the paper would benefit from tables that stand alone, with less commentary in the results section.
--

REVIEWER	Amy Huebschmann University of Colorado School of Medicine
REVIEW RETURNED	05-May-2014

- The reviewer completed the checklist but made no further comments.

VERSION 2 – AUTHOR RESPONSE

Reviewer Name Paul Froom
 Institution and Country school of Public health
 University Tel Aviv
 Please state any competing interests or state 'None declared': none

First of it would be beneficial to be able to read my previous review of this paper.
 Secondly, the regression models should have reported the areas under the curve.
 Thirdly, the paper would benefit from tables that stand alone, with less commentary in the results section.

It might be accepted as is, but could be improved.

Response: Thank you again for taking your time to review our paper. We suppose the first comment is directed to the editor. Secondly, we have discussed the issue of reporting the AUC in the paper further. There seems to be conflicting opinions on whether ROC analyses of model performance is necessary in association studies, but we have agreed to report the AUC as a description of the model classification performance.

Thirdly; some minor changes were made in the paper to avoid repeating figures from tables in the results section.

Reviewer Name Amy Huebschmann

Institution and Country University of Colorado School of Medicine

Please state any competing interests or state 'None declared': None declared

Authors addressed my initial concerns in this revision.

Response: Thank you again for your valuable initial comments, helping us to improve the quality of the paper.